# Heritability of temperature-mediated flower size plasticity in *Arabidopsis thaliana*

Gregory M. Andreou[1], Michaela Messer[2], Hao Tong[3,4], Zoran Nikoloski[3,4] and Roosa A. E. Laitinen[1,2] 

[1]Organismal and Evolutionary Research Programme, Faculty of Biological and Environmental Sciences, Viikki Plant Science Centre, University of Helsinki, Helsinki, Finland; [2]Molecular Mechanisms of Plant Adaptation Group, Max Planck Institute of Molecular Plant Physiology, Potsdam, Germany; [3]Bioinformatics Department, Institute of Biochemistry and Biology, University of Potsdam, Potsdam, Germany; [4]Systems Biology and Mathematical Modeling Group, Max Planck Institute of Molecular Plant Physiology, Potsdam, Germany

## Original Research Article

**Keywords:**
*Arabidopsis thaliana*; heritability; inheritance patterns; plasticity; temperature.

G.M.A. and M.M. contributed equally to this work.

**Author for correspondence:**
Roosa A. E. Laitinen,
E-mail: Roosa.Laitinen@helsinki.fi

### Abstract

Phenotypic plasticity is a heritable trait that provides sessile organisms a strategy to rapidly mitigate negative effects of environmental change. Yet, we have little understanding of the mode of inheritance and genetic architecture of plasticity in different focal traits relevant to agricultural applications. This study builds on our recent discovery of genes controlling temperature-mediated flower size plasticity in *Arabidopsis thaliana* and focuses on dissecting the mode of inheritance and combining ability of plasticity in the context of plant breeding. We created a full diallel cross using 12 *A. thaliana* accessions displaying different temperature-mediated flower size plasticities, scored as the fold change between two temperatures. Griffing's analysis of variance in flower size plasticity indicated that non-additive genetic action shapes this trait and pointed at challenges and opportunities when breeding for reduced plasticity. Our findings provide an outlook of flower size plasticity that is important for developing resilient crops for future climates.

## 1. Introduction

Plasticity denotes the ability of an organism to adjust its phenotype in response to an environmental change without modifying its' genotype (Bradshaw, 1965; Laitinen & Nikoloski, 2019; Pigliucci, 2005). As a result, plasticity allows sessile organisms, like plants, to adapt to new environments in temporal scales shorter than generation times (Bradshaw, 1965). Bradshaw (1965) proposed that plasticity of a focal trait to an environmental cue can be considered a trait that is under genetic control. Since then, several plasticity genes have been identified (Laitinen & Nikoloski, 2019). The genetic basis of plasticity makes it accessible to selection and thereby contributing to evolution (Bradshaw, 2006).

In addition to its evolutionary importance, understanding the genetic control of trait plasticity in plants in response to different environmental cues is essential in agricultural applications. For instance, lack of plasticity in yield, that is, exhibiting stable yield to varying field conditions is particularly relevant to meet human demand for food and feed. This is becoming especially important under more fluctuating conditions expected due to climate change. Furthermore, genome-wide association analysis has been applied to identify the genetic architecture and markers for yield stability to drought in soybean (Quero et al., 2021), trait stability grain yield, heading date and plant height in wheat (Lozada & Carter, 2020) and grain yield in barley (Ingvordsen et al., 2015). Beside the identification and characterization of plasticity genes, other equally important aspects of research in plasticity should include: (a) the quantification of components of genetic variance, allowing the determination of heritability, and (b) the characterization of the genetic components of plasticity explained by gene actions, including: additive, dominance, overdominance, and epistatic effects, relevant in breeding (Falconer & Mackay, 1996). Yet, despite the importance of plasticity, there are very few studies focused on addressing these two problems.

Following the classical model of quantitative genetics, the phenotypic variation of trait in response to a set of environments can be divided into genotype, environment, and

genotype-by-environment interaction variance components (Falconer & Mackay, 1996). Based on this, Scheiner and Lyman (1989) have defined the so-called plastic variance as the sum of the environment and genotype-by-environment interaction variance components. Genotype-by-environment interaction variance indicates that there is genetic variability for the response of genotypes to changes in the environment. Furthermore, the heritability of plasticity has been defined as the proportion of the genotype-by-environment interaction variance from the total phenotypic variance (Scheiner & Lyman, 1989). However, these analyses do not require quantification of plasticity of the focal trait but instead are solely based on scoring the mean of the focal trait in multiple genotypes exposed to different environments. In the case of two environments, Scheiner and Lyman (1989) have also shown that heritability of plasticity can be estimated by measuring plasticity using parent–offspring regression based on the difference in the focal trait expressions between the two environments. In the seminal study of Khan et al. (1976), it was shown that when plasticity of focal trait is scored by the fold change (FC; i.e., ratio) of the trait between two environments, heritability of plasticity can be as large as 0.5.

Since plasticity to an environmental cue can be seen as a heritable trait, its variance in a population of genotypes can also be decomposed using classical quantitative genetics models. However, this analysis requires scoring the plasticity in two or more environments. Among multiple ways to quantify plasticity (Laitinen & Nikoloski, 2019), FC between two environments is the simplest to measure, and differences in FCs between genotypes corresponds to the differences in slopes of the reaction norms of the focal traits. FCs have already been used to dissect the genetic architecture of plasticity of different focal traits in various species, including maize (Kusmec et al., 2017), barley (Long et al., 2019), and *A. thaliana* (Duarte et al., 2021; Meyer et al., 2019). Nevertheless, investigations of the variance components of plasticities are not usually carried out.

Breeding is based on generation of crosses from a population of parental genotypes. To this end, analysis of general or specific combining abilities can help in identifying superior parents to be used in breeding programs or to identify hybrids for cultivar development (Lynch & Walsh, 1998). General combining ability (GCA) directly relates to the breeding value of a parent and is associated with additive genetic effects, while specific combining ability (SCA) is the relative performance of a cross that is associated with non-additive gene action, for example, dominance, epistasis, or genotype-by-environment interaction (Falconer & Mackay, 1996). Therefore, both GCA and SCA effects are important in the selection or development of breeding populations. If we are interested in breeding for plasticity, or conversely robustness, it is necessary to determine GCA and SCA for plasticity of a focal trait.

In line with the analysis of combining ability, the mode of inheritance of plasticity can be characterized in a similar way to what is done with focal traits. For instance, lack of difference in plasticity from the mid-parent value (MPV) indicates additive mode of inheritance. Any deviation from the MPV confined in the range of parental plasticities would indicate dominance, and values outside of this range denote overdominance. Characterizing GCA, SCA, and the mode of inheritance for plasticity can be readily addressed by having accesses to $p$-scores from crosses obtained from parents with contrasting plasticities.

Flower size is one of the key traits defining the reproductive strategy of plants and therefore associated with reproductive success and fitness of a plant. Due to the direct involvement in fitness of the plant, flower size has thought to be maintained highly stable within one species. Hence, breeding for flower size robustness is particularly relevant for plant breeding. We have recently identified that *MAF2-5* gene cluster is associated with flower size plasticity to temperature based on genome-wide association with 290 *A. thaliana* accessions (Wiszniewski et al., 2022). In addition, mutant analysis showed that *MAF2-5* gene cluster is responsible for temperature-mediated flower size plasticity (Wiszniewski et al., 2022).

For crops, stability of flower and fruit size is desired, particularly in future climate scenarios, to obtain uniform and predictable yield. Therefore, to further investigate heritability of the temperature-mediated flower size plasticity in *A. thaliana,* we made a full diallel design of 12 accessions. The accessions were chosen to exhibit a range of temperature-induced flower plasticity. To examine the inheritance patterns of temperature-induced flower size plasticity, the classic quantitative design derived from Griffing (1956) was applied.

Our findings showed that flower size exhibits plasticity that is mostly showing additive inheritance. Hence, the plasticity of progeny can be directly predicted from the phenotype of the parents. Only 11 out of 134 crosses showed dominance from which one reciprocal cross showed overdominance. In all these cases, an increase in the amount of plasticity shown was dominantly inherited. These results provide insights in challenges of breeding for plasticity or robustness as dominance in addition to additive inheritance must be considered.

## 2. Methods

### 2.1. Plant material and growth conditions

Twelve parental accessions of *A. thaliana* expressing a range of temperature-mediated flower size plasticities were selected based on Wiszniewski et al. (2022). All plants were grown in individual 6 cm diameter pots and were first placed for 1 day at 20°C/6°C under long-day (LD) conditions 16 hr light:8 hr dark, with a photon flux density of 250 μM/m$^2$ 410/s. All pots were then vernalised at 4°C for at least 6 weeks. After vernalization, the plants were grown at either 17 or 23°C under LD conditions with photon flux 140 μM/m$^2$ 415/s and relative humidity ~70%. To control possible local effects of the growth chambers, the trays were moved and rotated every second day.

### 2.2. Trait measurements

Flower diameter (FD) was used a proxy for flower size (Wiszniewski et al., 2022). Two individuals of each parent (Table S1 in the Supplementary Material) were grown and crossed using a classical full diallel crossing design resulting in two sets of diallel crosses. Each parent was also crossed to itself to avoid any effect due to manual fertilization. The first filial (F$_1$) hybrids were genotyped to ensure heterozygosity (primers listed in Table S2 in the Supplementary Material). These two sets represent the biological replicates grown in two separate trials. To record FDs for each of the individual plants, at least six open flowers from the primary inflorescence were collected, resulting in at least 12 replicates for every genotype. To minimize the technical variation, flowers were only taken after the eighth flower had opened and were always harvested in the morning after the lights had been on for at least 3 hr. Flowers were placed on a 96-well plate containing 1–2% agarose and macrographs were taken. FDs as two diagonals

were measured from macrographs, using imageJ software, as an average of all measurements. To quantify plasticity of flower size to temperature, the FC between 23 and 17°C was calculated. For each genotype, four independent FCs were calculated by dividing each of the two mean FDs of the two biological replicates at 23°C with each of the two mean FDs of the two biological replicates at 17°C.

### 2.3. Mode of inheritance

To identify significant differences between the flower size plasticities in hybrids and their respective MPVs, one sample two-tailed *t*-tests were performed. *P*-values were adjusted according to the Benjamini–Hochberg procedure for multiple hypotheses test correction, and a false discovery rate of 10%. To determine dominance or overdominance, hybrids that showed a significant difference in flower size plasticity to their MPV were then tested against the plasticity of their mother and father using two-sample two-tailed *t*-tests assuming equal or non-equal variances, based on the result of the preceding *F*-tests. The resulting *p*-values were adjusted, while allowing a false discovery rate of 10%.

### 2.4. Analysis of combining ability

Here, we considered Method I of Griffing's (1956) analysis of combining ability, with all general and specific combining abilities considered as fixed effects. We considered fixed effects only since we were interested in estimates of GCA and SCA for the specific parents and crosses included in the design. This analysis was performed using the lmDiallel package in R (Onofri et al., 2021). Heritability of plasticity, scored as FC, can be determined by using the calculation of variance in GCA and SCA from full diallel crossing design, since the additive variance is four times the variance in GCA and the dominance variance is four times the variance in SCA. This estimate of heritability was further supported with father–offspring regression estimate.

To detect significant differences between overall parent and hybrid genotypes, a Wilcoxon ranked sum test was employed to compare flower size plasticities from the hybrids and the parents, using the wilcox_test() function in the "rstatix" package. Levene's test for homogeneity of variances between the two groups compared was also carried out, using the leveneTest() function in the "car" package.

## 3. Results and discussion

### 3.1. Full diallel design to study temperature-mediated flower size plasticity in A. thaliana

To obtain insights in the mode of inheritance of temperature-mediated flower size plasticity in *A. thaliana*, we generated a full diallel crossings with 12 natural accessions (Table S1 in the Supplementary Material). Flower sizes for all crosses were scored at 23 and at 17°C with two biological replicates (see Section 2). Here, we used the FC between the flower size at 23 and 17°C as a measure for temperature-mediated flower size plasticity, allowing us to obtain four replicates of FCs for each cross (see Section 2). The temperature-mediated flower size plasticity in these accessions ranged from 0.79 to 1.10 (Figure 1 and Figure S1 in the Supplementary Material), which includes the mean average flower size plasticity of 0.86 observed in the larger panel of *A. thaliana* accessions used to identify the plasticity genes underlying this trait (Wiszniewski et al., 2022). From the 12 accessions used in this

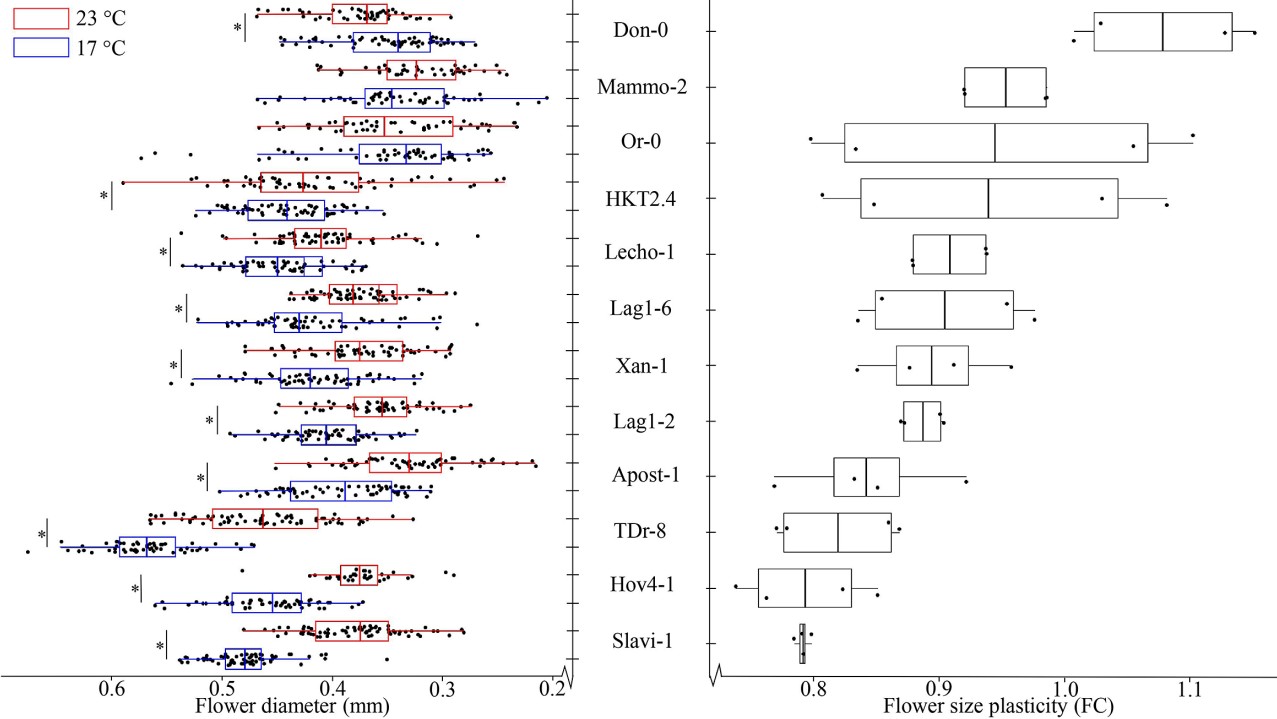

**Fig. 1.** Mean average flower diameters and temperature-mediated flower size plasticities in the parental accessions used for diallel crosses. Flower diameters were measured at each temperature and compared using a two-sample two-tailed *t*-test. ∗ indicates *p*-values < .1, corrected by Benjamini–Hochberg when *n* = 12. Flower size plasticity is quantified as the ratio between the two biological replicates of plants grown at 23 and 17°C. Accessions are ordered from smallest to largest average temperature-mediated flower size plasticity. Error bars represent standard deviation of the mean for the FDs and the plasticities measured.

study, 10 exhibited significant differences between the mean values of flower size between 23 and 17°C (two-sample two-tailed *t*-test, adjusting the *p*-value by the Benjamini–Hochberg method; *n* = 12, FDR = 10%; Figure 1). Slavianka-1 (Slavi-1, Bulgaria) and Hov4-1 (Hov4-1, Sweden) showed the smallest average FCs, which was however different from one. Thus, these accessions exhibited the largest plasticity in flower size. Contrastingly, two of the accessions, Mammola-2 (Mammo-2, Italy) and Oranienstein-0 (Or-0, Germany), showed FCs close to one and lacked significant difference in the means of their flower size between the two temperatures, suggesting that they did not exhibit any plasticity (*p*-value > .1). In addition to these two accessions, Donana-0 (Don-0, Spain) also had FC close to one, implying lack of plasticity; in this accession, there was a significant but small difference between the mean values of flower sizes between 23 and 17°C (3.76 mm and 3.49 mm, respectively; *p*-value < .1, Figure 1). We note that values of FC closer to one would be preferred for plant breeding, since they do not show plasticity, that is, they remain stable at the two temperatures used in the experiments.

### 3.2. Mode of inheritance of temperature-mediated flower size plasticity

To characterize the most frequent mode of inheritance, we calculated the MPVs and investigated where the mean of temperature-mediated flower size plasticity of each cross is positioned with respect to the range of plasticities for the parents. By conducting one-sample *t*-tests, corrected for multiple hypotheses tests employing the Benjamini–Hochberg procedure, we identified 14 crosses that exhibited FC significantly different from the MPV at false discovery rate of 10%. This result indicated non-additive inheritance

in these crosses (Table S3 in the Supplementary Material). Hence, the most common mode of inheritance observed was additive inheritance, as the other 118 crosses did not show significantly different FC in comparison to their MPV.

Next, we investigated how many of the 14 non-additive cases were due to dominance or overdominance. We found that 11 out of the 14 hybrids only showed significant different FC from one of the parents and thereby exhibited full dominance (Figure 2). In all these crosses, reduced FC was present, implying greater capacity for plasticity. From these 11, in six and three cases, the mother and the father, respectively, were the dominant parent suggesting that there is no parental effect in the inheritance of temperature-mediated flower size plasticity (Figure 2a). Moreover, two (Lag1-2 x Lecho-1 and Lecho-1 × Lag1-2) out of the 11 crosses showed significantly different FC from both parents, implying overdominance (Figure 2b). Both crosses showed higher plasticity (smaller FC) than either of the parents. The final three out of the 14 crosses did not show significant differences in mean average flower size plasticity to either parent, despite showing a significant difference in temperature-mediated flower size plasticity from MPV (Figure S2 in the Supplementary Material). It is possible that significant differences were not found between these three crosses and their respective parent's average flower size plasticity FC as the standard deviation around the mean for the parents overlapped. Nevertheless, these three cases were also examples of negative dominance, as the FC in temperature-mediated flower size plasticity for the hybrid was lower than that of the MPV.

Altogether, these results demonstrate that in most cases the inheritance patterns of the temperature-mediated flower size plasticity were additive, allowing prediction from the phenotype of the parents. Nevertheless, in the rare case of non-additive inheritance,

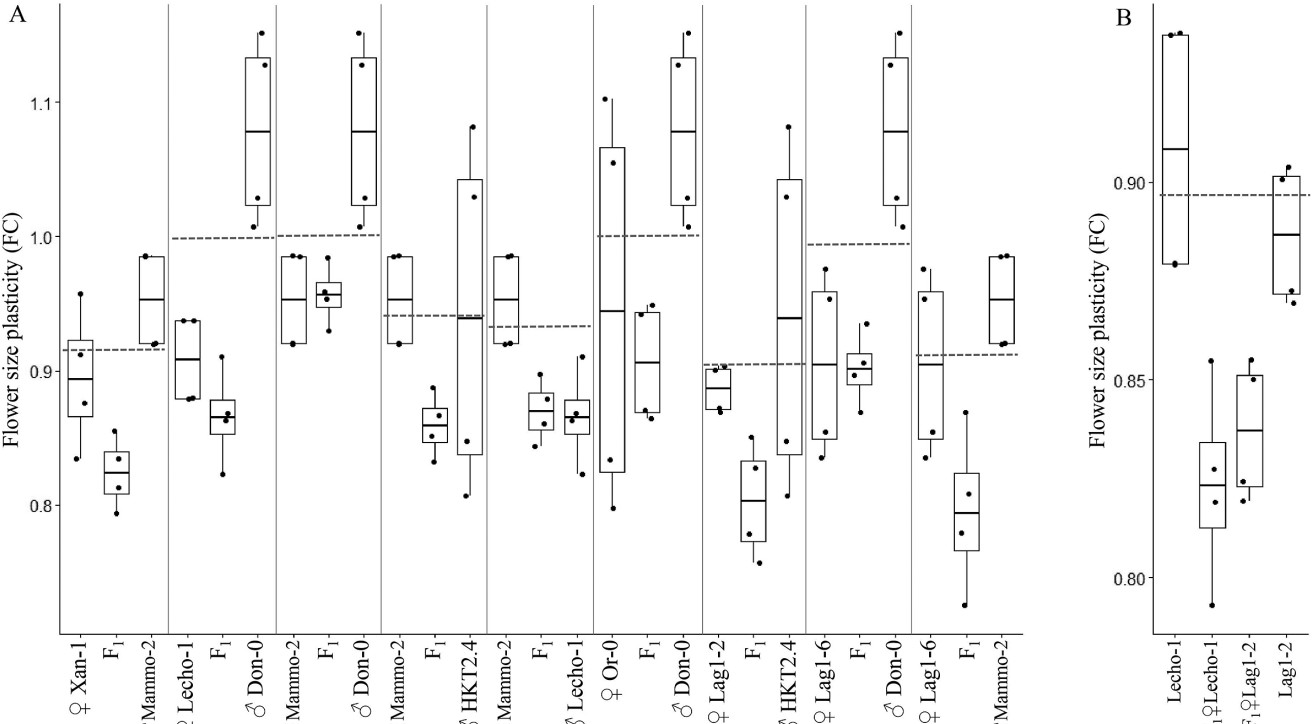

**Fig. 2.** Crosses showing non-additive inheritance of temperature-mediated flower size plasticity. All crosses shown exhibited a significant difference in calculated FC plasticity compared to their respective MPV at a false discovery rate of 10% (shown by the dashed line, *p*-values were adjusted according to the Benjamini–Hochberg, when *n* = 28. All crosses showed negative dominance. (a) Six showed dominance towards maternal and three showed dominance towards parental alleles. The reciprocal crosses between Lag1-2 and Lecho-1 showed negative overdominance (b), whereby the average temperature-mediated flower size plasticity FC was significantly greater than both parental phenotypes.

**Table 1.** Variances for the general and specific combining ability effects of temperature-mediated flower size plasticity.

| | Variance | SE | Df | Sum sq. | Mean sq. | F-value | Pr(>F) |
|---|---|---|---|---|---|---|---|
| GCA | 6.39E−04 | 3.39E−04 | 11 | 0.87 | 0.08 | 13.16 | 1.45E-21 |
| tSCA | 1.24E−03 | 3.72E−04 | 66 | 1.00 | 0.02 | 2.52 | 1.61E-08 |
| Reciprocals | 1.02E−03 | 3.20E−04 | 65 | 0.94 | 0.01 | 2.42 | 8.22E-08 |
| Residuals | 6.07E−03 | 4.20E−04 | 414 | 2.48 | NA | NA | NA |

*Note*: Analysis was performed using full diallel experiment with 12 accessions of *A. thaliana* using Griffing's Method 1. The table includes the significance of general (GCA), total specific combining abilities (tSCA), as well as reciprocal effects.

the increased plasticity alleles are dominantly inherited. The presence of non-additive inheritance cannot be captured with the most widely used methods used to predict the estimated breeding values in agriculture and could have dramatic effect on breeding in which the additive inheritance is preferred (Varona et al., 2018). Yet, to find out if the observed additive inheritance of flower size plasticity to temperature applies to other trait plasticities and different environmental cues remains to be investigated in future.

### 3.3. General and specific combining abilities

To analyse the combining ability, we next employed the temperature-mediated flower size plasticities from our diallel design in Griffing's Method 1 (Griffing, 1956; Section 2). This analysis is suitable since we include all crosses, and reciprocals, along with the parents. We used fixed effects in the implementation of the model, since we did not aim to make inferences to other populations. Accession Lechovo-1 (Lecho-1, Romania) had to be removed from the analyses, since it caused numerical instability issues due to singularities in the data set.

First, we examined the variance components of the GCA and SCA and found that the variance of GCA is an order or magnitude smaller than the variance of SCA (Table 1). As a result, we concluded that the dominance variance is an order of magnitude larger than the additive variance, suggesting that non-additive gene action shapes temperature-mediated flower size plasticity. In support of this claim, the general predicted ratio (0.91) is larger than 0.5, calculated as 2∗mean squares − GCA/(2∗mean squares − GCA + mean squares − SCA) (Table 1) which also suggested non-additive genetic effects (Baker, 1978).

Three of the accessions showed significant GCA at threshold of 0.01, including Don-0, TDr-8, and Hov4-1 (Table 2). Of these, two exhibited negative, namely TDr-8 and Hov4-1, GCA that are in the direction of selection for reduced plasticity. Altogether, we found that five of the 11 accessions assessed showed positive GCA, which led to an increase in plasticity when used as parents in crosses. Of these, only one accession, namely HKT2.4, showed significant positive GCA. In plant breeding, reduced plasticity of desired trait in crosses would be desired while increase in plasticity is unwanted.

With respect to SCA, we found that only 11 crosses exhibited significant SCA indicative of a non-additive inheritance (Table S3 in the Supplementary Material). From these, only two crosses (Lag1-2 × HKT2.4 and Mammo-2 × Xan-1) also showed non-additive inheritance in our previous analysis. This refers to the fact that SCA is a relative analysis in the context of the analysed population. The smallest negative SCA was observed for the Or-0 × Lag1-2 cross.

Finally, using father–offspring regression with the assembled data we found that the heritability of the temperature-mediated

**Table 2.** General combining abilities of the parental accessions that were used for this study.

| Accession name | GCA estimate | SE | T-value | Pr(> \|t\|) |
|---|---|---|---|---|
| Apost-1 | −0.02 | 0.01 | −2.35 | 1.94E-02 |
| Don-0 | 0.08 | 0.01 | 9.02 | 6.99E-18 |
| HKT2.4 | 0.01 | 0.01 | 1.45 | 1.49E-01 |
| Hov4-1 | −0.03 | 0.01 | −3.91 | 1.07E-04 |
| Lag1-2 | −0.01 | 0.01 | −1.23 | 2.21E-01 |
| Lag1-6 | −0.01 | 0.01 | −0.68 | 5.00E-01 |
| Mammo-2 | 0.00 | 0.01 | 0.62 | 5.37E-01 |
| Or-0 | 0.01 | 0.01 | 0.67 | 5.04E-01 |
| Slavi-1 | −0.01 | 0.01 | −1.81 | 7.03E-02 |
| TDr-8 | −0.03 | 0.01 | −3.90 | 1.14E-04 |
| Xan-1 | 0.02 | 0.01 | 2.20 | 2.84E-02 |

*Note*: Griffing's method was applied for general combining ability in the analysed population.

flower size plasticity is not larger than was 0.43 (i.e., twice the regression coefficient, Figure 3a). This was supported by an estimate of heritability using the variance components of GCA and SCA, whereby we found that the heritability of temperature-induced flower size plasticity was 0.67 (see Section 2). The discrepancy in the estimates is in part due to the exclusion of one genotypes from the analysis of combining abilities. Therefore, heritability in plasticity is sizeable, but is in the order of other traits related to yield and fitness in *A. thaliana* (Seymour et al., 2016).

These findings suggest that temperature-mediated flower size plasticity has a non-additive genetic component found by analyses of combining abilities as well as mode of inheritance. They also highlight the importance of assessing the heritability of genotypes used for breeding to identify those associated with desired high stability.

### 3.4. Relationship between plasticity and heterozygosity

The diallel design allowed us also to ask if the temperature-mediated flower size plasticity is associated with heterozygosity of the genotypes. We have previously shown that outcrossing species with increased complexity due to heterozygosity show less plasticity than highly homozygous selfing species (Wiszniewski et al., 2022). On the other hand, crosses among the different alleles for high plasticity and low plasticity could add to the genetic combinations and increase the range of plasticity values due to non-additive inheritance and epistasis and increase the plasticity in hybrids. To test if heterozygosity increases plasticity, which has been already suggested (see Section 1), we grouped

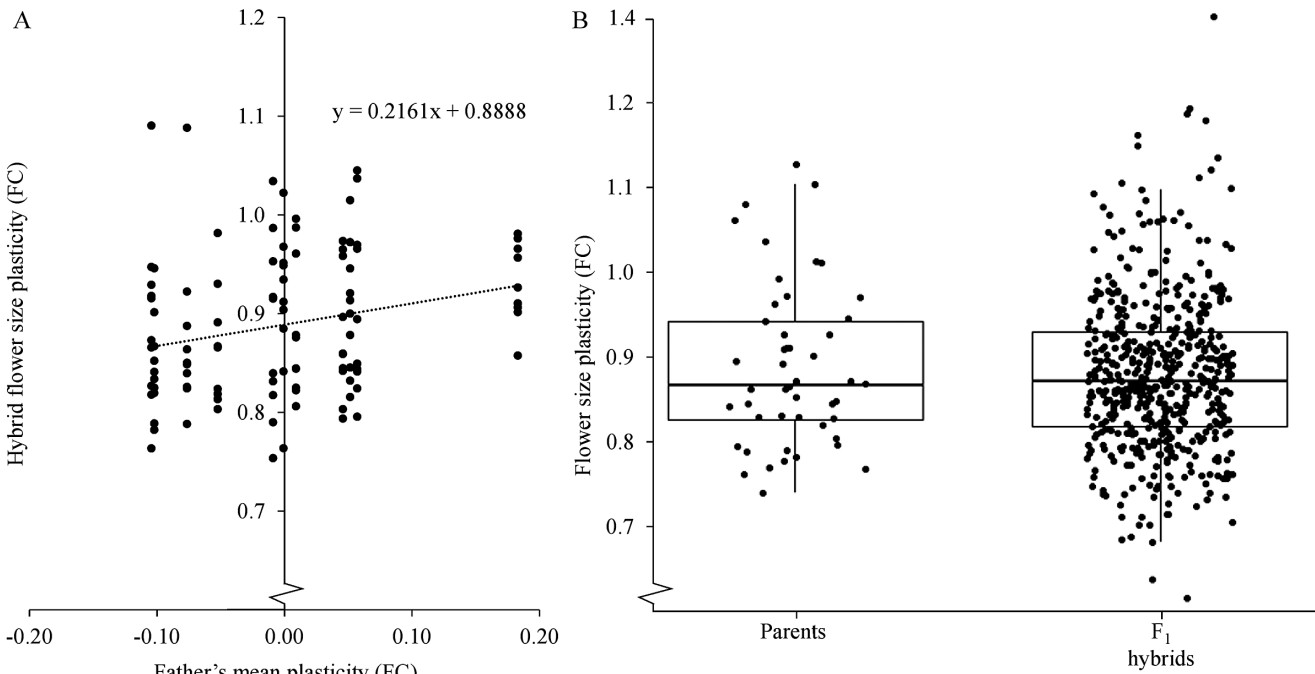

**Fig. 3.** Heritability and box plots for flower diameter plasticities of parent and F1 hybrids. (a) Father–offspring regression to quantify heritability (i.e., twice the regression coefficient). (b)Wilcoxon ranked rum test applied showing no significant differences between the groups ($W = 690$, $p$-value = .0487). Levene's test for homogeneity suggests the variances cannot be distinguished between the two groups ($F = 0.1415$, $p$-value = .7074).

all hybrids and analysed if they exhibited significantly different average mean plasticity values than the homozygous parents. While the heterozygotes (i.e., hybrids) showed slightly smaller plasticity than the parents, the difference between plasticities in these two groups were non-significant (Figure 3b). Furthermore, consistent with our earlier observation that most of the hybrids showed additive inheritance patterns of temperature-mediated flower size plasticity. Furthermore, in line with our finding of cases of non-additive inheritance in the hybrids, we observed a larger spread of the plasticity values in hybrids than in parents (Figure 3b).

To conclude, our findings indicate that although with high potential, breeding of plasticity is challenging due to non-additive inheritance and low heritability. Our results highlight the importance to select the suitable genotypes to breeding for plasticity and profoundly increase understanding of the impact of temperature on phenotypic plasticity, and the challenges it imposes in plant species.

## Acknowledgements

We thank Karin Köhl and the Green Team of the Max Planck Institute of Molecular Plant Physiology for taking excellent care of plants. Joona Huotari is thanked for his illustrations in graphical abstract. Open access funded by Helsinki University Library.

**Financial support.** This research received no specific grant from any funding agency, commercial or not-for-profit sectors.

**Conflict of interest.** The authors declare no conflicts of interest.

**Authorship contributions.** R.A.E.L. designed the experiments; M.M. performed the experiments; G.A. and H.T. analysed the data; G.A. prepared the figures; Z.N. and R.A.E.L. supervised the experiments; G.A., Z.N., and R.A.E.L. wrote the manuscript.

**Data availability statement.** The raw data supporting the conclusions of this article will be made available by the authors, without undue reservation, to any qualified researcher.

**Supplementary materials.** To view supplementary material for this article, please visit http://doi.org/10.1017/qpb.2023.3.

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
