## [Reviewer Report · Author comment: Heritability of temperature-mediated flower size plasticity in *Arabidopsis thaliana*— R1/PR4]

Dear Prof. Hamant,

We would like to submit manuscript titled “Heritability of temperature-mediated flower size plasticity in *Arabidopsis thaliana*” to be considered as a research article in Quantitative Plant Biology.

Phenotypic plasticity denotes the ability of a genotype to modify its phenotypes in response to change in environment. It is particularly important for plants, which as sessile organisms need to quickly mitigate negative effects of environmental changes on development, growth, and reproduction. Temperature is one of the main indicators of climate change, and many traits (including flower size) are known to induce response to an increase in temperature. We have recently investigates genetic basis of temperature-mediated flower size plasticity. In our recent wrok (Wiszniewski et al. 2022), using genome-wide analysis combined with characterization of mutant lines, we showed that temperature-mediated is associated with MAF2-5 locus on chromosome 5.

This study builds on this discovery to further investigate the full potential of temperature-mediated flower size plasticity, especially in the context of plant breeding. To this end, we have made full diallel crosses among twelwe parents showing different flower size plasticities and assessed the heritability patterns of this focal trait. Our findings showed that:

- Overall the heritability of temperature-mediated flower size plasticity was moderate, but comparable to that of yield and fitness in *A. thaliana*.

- While majority of the first generation crosses showed additive inheritance, with only 11 out of 242 accessions that showed non-additive inheritance.

- The non-additively inherited crosses all showed dominance towards plasticity increase.

- Plasticity in heterozygous hybrids exhibited a larger range of values than in homozygous parents, but heterozygous plants did not show more plasticity than homozygous parents.

To conclude, our findings indicate that although with high potential, breeding of plasticity is challenging due to non-additive inheritance and low heritability. Our results highlights the importance of the suitable genotypes to breeding for plasticity and profoundly increase understanding of the impact of temperature on phenotypic plasticity, and the challenges it imposes in plant species.

Sincerely,

Roosa Laitinen

---

## [Reviewer Report · Review: Heritability of temperature-mediated flower size plasticity in *Arabidopsis thaliana*— R1/PR5]

*Comments to Author*: This manuscript addresses the genetic basis of phenotypic plasticity. The phenotype studied is Arabidopsis flower size, which shows plasticity to ambient temperature. This study builds on a recent publication of a gene cluster associated with this plasticity. Therefore, the description of the genetic inheritance of plasticity presented here is timely and important.

A panel of 12 accessions were used that represent natural variation for this plasticity. The majority of accessions showed temperature-mediated flower size plasticity (scored as fold change between 23’C and 17’C), while 3 accessions lacked plasticity. The authors created a full diallel cross with the 12 accessions and analysed mode of inheritance and combining ability. These are standard methods in quantitative genetics and plant breeding. Although this is not my area of expertise, I believe that the contributing authors have the appropriate expertise to design and analyse these experiments correctly.

The authors could estimate the heritability of temperature-mediated flower size plasticity and demonstrate that it has mostly additive, but some non-additive inheritance. They conclude that low heritability and non-additivity make breeding for plasticity challenging. These are important findings that increase our understanding of the genetic basis of phenotypic plasticity. I was left with the impression that the authors were discussing breeding in Arabidopsis at some point (!), so they could check that they make a clear distinction between how their findings can be generally useful for crop breeding (where uniformity is desired), versus how their findings are useful to understand the specific trait under study in Arabidopsis.

---

## [Reviewer Report · Review: Heritability of temperature-mediated flower size plasticity in *Arabidopsis thaliana*— R1/PR6]

*Comments to Author*: In this article the authors analyzed the heritability of temperature-mediated flower size plasticity in Arabidopsis. They showed that among 134 crosses using different parental lines, most showed additive inheritance in flower size plasticity, and only 11 crosses showed dominance inheritance. The findings are interesting and provide evidence of flower size plasticity for breeding.

1. Mammo-2 and Or-0 did not exhibit any plasticity, while some accessions showed large plasticity. Is there any possible link between temperature-mediated flower size plasticity and the growth environment of the original place of the accessions?

2. The results showed that in most cases the inheritance patterns of the temperature-mediated flower size plasticity were additive. Is this similar with the inheritance patterns of plasticity of traits, for instance flowering time or yield？This can be included in the discussion.

3. In fig 2 and fig 3, the labels are too small to read clearly.